# Deep Hierarchical Planning from Pixels

**Danijar Hafner** [1 2 3]    **Kuang-Huei Lee** [2]    **Ian Fischer** [2]    **Pieter Abbeel** [1 4]

## Abstract

Intelligent agents need to select long sequences of actions to solve complex tasks. While humans easily break down tasks into subgoals and reach them through millions of muscle commands, current artificial intelligence is limited to tasks with horizons of a few hundred decisions, despite large compute budgets. Research on hierarchical reinforcement learning aims to overcome this limitation but has proven to be challenging, current methods rely on manually specified goal spaces or subtasks, and no general solution exists. We introduce Director, a practical method for learning hierarchical behaviors directly from pixels by planning inside the latent space of a learned world model. The high-level policy maximizes task and exploration rewards by selecting latent goals and the low-level policy learns to achieve the goals. Despite operating in latent space, the decisions are interpretable because the world model can decode goals into images for visualization. Director outperforms exploration methods on tasks with sparse rewards, including 3D maze traversal with a quadruped robot from an egocentric camera and proprioception, without access to the global position or top-down view that was used by prior work. Director also learns successful behaviors across a wide range of environments, including visual control, Atari games, and DMLab levels.

## 1 Introduction

Artificial agents have achieved remarkable performance on reactive video games (Mnih et al., 2015; Badia et al., 2020) or board games that last for a few hundred moves (Silver et al., 2017). However, solving complex control problems can require millions of time steps. For example, consider a robot that needs to navigate along the sidewalk and cross streets to buy groceries and then return home and

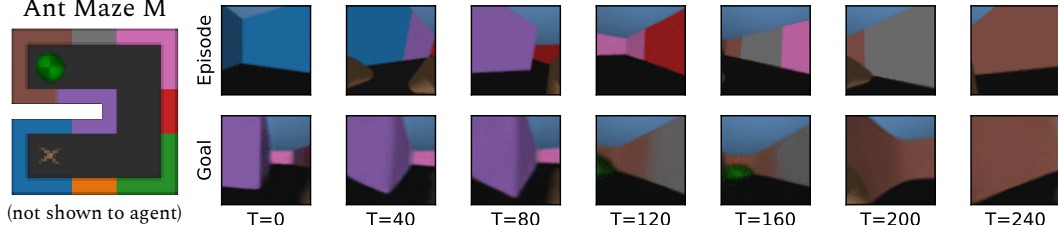

**Figure 1:** Director on Ant Maze M from egocentric camera inputs. The top row shows agent inputs. The bottom row shows the internal subgoals of the agent. The goals are latent vectors that Director's world model can decode into images for human inspection. Director solves this sparse reward task by breaking it down into internal goals. It first targets the purple wall in the middle of the maze. Once reached, it targets the reward object, and then the brown wall behind it to step onto the reward object.

---

[1]UC Berkeley   [2]Google Research   [3]University of Toronto   [4]Covariant

Correspondence to: Danijar Hafner <mail@danijar.com>

Project website with videos and code: https://danijar.com/director

36th Conference on Neural Information Processing Systems (NeurIPS 2022).

cook a meal with those groceries. Manually specifying subtasks or dense rewards for such complex tasks would not only be expensive but also prone to errors and require tremendous effort to capture special cases (Chen et al., 2021; Ahn et al., 2022). Even training a robot to simply walk forward can require specifying ten different reward terms (Kumar et al., 2021), making reward engineering a critical component of such systems. Humans naturally break long tasks into subgoals, each of which is easy to achieve. In contrast, most current reinforcement learning algorithms reason purely at the clock rate of their primitive actions. This poses a key bottleneck of current reinforcement learning methods that could be challenging to solve by simply increasing the computational budget.

Hierarchical reinforcement learning (HRL) (Dayan and Hinton, 1992; Parr and Russell, 1997; Sutton et al., 1999) aims to automatically break long-horizon tasks into subgoals or commands that are easier to achieve, typically by learning high-level controllers that operate at more abstract time scales and provide commands to low-level controllers that select primitive actions. However, most HRL approaches require domain knowledge to break down tasks, either through manually specified subtasks (Tessler et al., 2017) or semantic goal spaces such as global XY coordinates for navigation tasks (Andrychowicz et al., 2017; Nachum et al., 2018a) or robot poses (Gehring et al., 2021). Attempts at learning hierarchies directly from sparse rewards have had limited success (Vezhnevets et al., 2017) and required providing task reward to the low-level controller, calling into question the benefit of their high-level controller.

In this paper, we present Director, a practical method for learning hierarchical behaviors directly from pixels by planning inside the latent space of a learned world model. We observe the effectiveness of Director on long-horizon tasks with very sparse rewards and demonstrate its generality by learning successfully in a wide range of domains. The key insights of Director are to leverage the representations of the world model, select goals in a compact discrete space to aid learning for the high-level policy, and to use a simple form of temporally-extended exploration in the high-level policy.

**Contributions** The key contributions of this paper are summarized as follows:

- We describe a practical, general, and interpretable algorithm for learning hierarchical behaviors within a world model trained from pixels, which we call Director (Section 2).
- We introduce two sparse reward benchmarks that underscore the limitations of traditional flat RL approaches and find that Director solves these challenging tasks (Section 3.1).
- We demonstrate that Director successfully learns in a wide range of traditional RL environments, including Atari, Control Suite, DMLab, and Crafter (Section 3.2).
- We visualize the latent goals that Director selects for breaking down various tasks, providing insights into its decision making (Section 3.3).

## 2 Director

Director is a reinforcement learning algorithm that learns hierarchical behaviors directly from pixels. As shown in Figure 2, Director learns a world model for representation learning and planning, a goal autoencoder that discretizes the possible goals to make them easier for the manager to choose, a manager policy that selects goals every fixed number of steps to maximize task and exploration rewards, and a worker policy that learns to reach the goals through primitive actions. All components are optimized throughout the course of learning by performing one gradient step every fixed number of environment steps. The world model is trained from a replay buffer whereas the goal autoencoder is trained on the world model representations and the policies are optimized from imagined rollouts. For the pseudo code of Director, refer to Appendix E.

### 2.1 World Model

Director learns a world model that compresses the history of observations into a compact feature space and enables planning in this space (Watter et al., 2015; Zhang et al., 2019). We use the Recurrent State Space Model (RSSM) model of PlaNet (Hafner et al., 2018), which we briefly review here to introduce notation. The world model consists of four neural networks that are optimized jointly:

$$\begin{array}{llll} \text{Model representation:} & \text{repr}_\theta(s_t \mid s_{t-1}, a_{t-1}, x_t) & \text{Model decoder:} & \text{rec}_\theta(s_t) \approx x_t \\ \text{Model dynamics:} & \text{dyn}_\theta(s_t \mid s_{t-1}, a_{t-1}) & \text{Reward predictor:} & \text{rew}_\theta(s_{t+1}) \approx r_t \end{array} \quad (1)$$

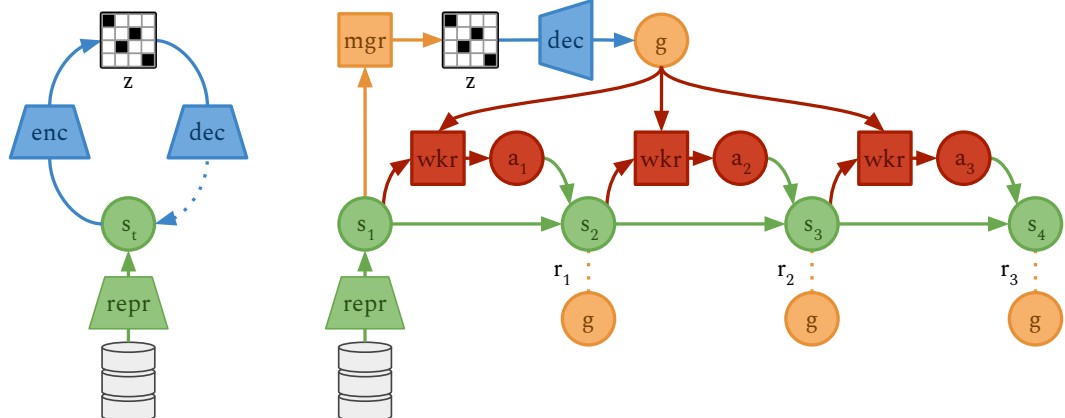

**Figure 2:** Director is based on the **world model** of PlaNet (Hafner et al., 2018) that predicts ahead in a compact representation space. The world model is trained by reconstructing images using a neural network not shown in the figure. Director then learns three additional components, which are all optimized concurrently. On the left, the **goal autoencoder** compresses the feature vectors $s_t$ into vectors of discrete codes $z \sim \text{enc}(z \mid s_t)$. On the right, the **manager** policy $\text{mgr}(z \mid s_t)$ selects abstract actions in this discrete space every $K = 8$ steps, which the goal decoder then turns into feature space goals $g = \text{dec}(z)$. The **worker** policy $\text{wkr}(a_t \mid s_t, g)$ receives the current feature vector and goal as input to decide primitive actions that maximize the similarity rewards $r_t$ to the goal. The manager maximizes the task reward and an exploration bonus based on the autoencoder reconstruction error, implementing temporally-extended exploration.

The representation model integrates actions $a_t$ and observations $x_t$ into the latent states $s_t$. The dynamics model predicts future states without the corresponding observations. The decoder reconstructs observations to provide a rich learning signal. The reward predictor later allows learning policies by planning in the compact latent space, without decoding images. The world model is optimized end-to-end on subsequences from the replay buffer by stochastic gradient descent on the variational objective (Hinton and Van Camp, 1993; Kingma and Welling, 2013; Rezende et al., 2014):

$$\mathcal{L}(\theta) \doteq \sum_{t=1}^{T} \Big( \beta \, \text{KL} \big[ \text{repr}_\theta(s_t \mid s_{t-1}, a_{t-1}, x_t) \, \big\| \, \text{dyn}_\theta(s_t \mid s_{t-1}, a_{t-1}) \big]$$
$$+ \| \text{rec}_\theta(s_t) - x_t \|^2 + (\text{rew}_\theta(s_{t+1}) - r_t)^2 \Big) \quad \text{where} \quad s_{1:T} \sim \text{repr}_\theta \tag{2}$$

The variational objective encourages learning a Markovian sequence of latent states with the following properties: The states should be informative of the corresponding observations and rewards, the dynamics model should predict future states accurately, and the representations should be formed such that they are easy to predict. The hyperparameter $\beta$ trades off the predictability of the latent states with the reconstruction quality (Beattie et al., 2016; Alemi et al., 2018).

## 2.2 Goal Autoencoder

The world model representations $s_t$ are 1024-dimensional continuous vectors. Selecting such representations as goals would be challenging for the manager because this constitutes a very high-dimensional continuous action space. To avoid a high-dimensional continuous control problem for the manager, Director compresses the representations $s_t$ into smaller discrete codes $z$ using a goal autoencoder that is trained on replay buffer model states from Equation 2:

$$\text{Goal Encoder:} \quad \text{enc}_\phi(z \mid s_t) \qquad \text{Goal Decoder:} \quad \text{dec}_\phi(z) \approx s_t \tag{3}$$

Simply representing each model state $s_t$ by a class in one large categorical vector would require roughly one category per distinct state in the environment. It would also prevent the manager from generalizing between its different outputs. Therefore, we opt for a factorized representation of multiple categoricals. Specifically, we choose the vector of categoricals approach introduced in DreamerV2 (Hafner et al., 2020a). As visualized in Figure G.1, the goal encoder takes a model state as input and predicts a matrix of 8×8 logits, samples a one-hot vector from each row, and flattens the results into a sparse vector with 8 out of 64 dimensions set to 1 and the others to 0. Gradients are

backpropagated through the sampling by straight-through estimation (Bengio et al., 2013). The goal autoencoder is optimized end-to-end by gradient descent on the variational objective:

$$\mathcal{L}(\phi) \doteq \left\| \text{dec}_\phi(z) - s_t \right\|^2 + \beta \, \text{KL}\left[ \text{enc}_\phi(z \mid s_t) \, \| \, p(z) \right] \quad \text{where} \quad z \sim \text{enc}_\phi(z \mid s_t) \tag{4}$$

The first term is a mean squared error that encourages the encoder to compute informative codes from which the input can be reconstructed. The second term encourages the encoder to use all available codes by regularizing the distribution towards a uniform prior $p(z)$. The autoencoder is trained at the same time as the world model but does not contribute gradients to the world model.

## 2.3 Manager Policy

Director learns a manager policy that selects a new goal for the worker every fixed number of $K = 8$ time steps. The manager is free to choose goals that are much further than 8 steps away from the current state, and in practice, it often learns to choose the most distant goals that the worker is able to achieve. Instead of selecting goals in the high-dimensional continuous latent space of the world model, the manager outputs abstract actions in the discrete code space of the goal autoencoder (Section 2.2). The manager actions are then decoded into world model representations before they are passed on to the worker as goals. To select actions in the code space, the manager outputs a vector of categorical distributions, analogous to the goal encoder in Section 2.2:

$$\text{Manager Policy:} \quad \text{mgr}_\psi(z \mid s_t) \tag{5}$$

The objective for the manager is to maximize the discounted sum of future task rewards and exploration rewards. The exploration encourages the manager to choose novel goals for the worker, resulting in temporally-abstract exploration. This is important because the worker is goal-conditioned, so without an explicit drive to expand the state distribution, it could prefer going back to previously common states it has been trained on the most, and thus hinder exploration of new states in the environment. Because the goal autoencoder is trained from the replay buffer, it tracks the current state distribution of the agent and we can reward novel states as those that have a high reconstruction error under the goal autoencoder:

$$r_t^{\text{expl}} \doteq \left\| \text{dec}_\phi(z) - s_{t+1} \right\|^2 \quad \text{where} \quad z \sim \text{enc}_\phi(z \mid s_{t+1}) \tag{6}$$

Both manager and worker policies are trained from the same imagined rollouts and optimized using Dreamer (Hafner et al., 2019; 2020a), which we summarize in Appendix H. The manager learns two state-value critics for the extrinsic and exploration rewards, respectively. The critics are used to fill in rewards beyond the imagination horizon and as baseline for variance reduction (Williams, 1992). We normalize the extrinsic and exploration returns by their exponential moving standard deviations before summing them with weights $w^{\text{extr}} = 1.0$ and $w^{\text{expl}} = 0.1$. For updating the manager, the imagined trajectory is temporally abstracted by selecting every $K$-th model state and by summing rewards within each non-overlapping subsequence of length $K$. No off-policy correction (Schulman et al., 2017; Nachum et al., 2018a) is needed because the imagined rollouts are on-policy.

## 2.4 Worker Policy

The worker is responsible for reaching the goals chosen by the manager. Because the manager outputs codes $z$ in the discrete space of the goal autoencoder, we first decode the goals into the state space of the world model $g \doteq \text{dec}(z)$. Conditioning the worker on decoded goals rather than the discrete codes has the benefit that its learning becomes approximately decoupled from the goal autoencoder. The worker policy is conditioned on the current state and goal, which changes every $K = 8$ time steps, and it produces primitive actions $a_t$ to reach the feature space goal:

$$\text{Worker Policy:} \quad \text{wkr}_\xi(a_t \mid s_t, g) \tag{7}$$

To reach its latent goals, we need to choose a reward function for the worker that measures the similarity between the current state $s_t$ and the current goal $g$, both of which are 1024-dimensional continuous activation vectors. Natural choices would be the negative L2 distance or the cosine similarity and their choice depends on the underlying feature space, which is difficult to reason about. Empirically, we found the cosine similarity to perform better. Cosine similarity would usually normalize both vectors, but normalizing the state vector encourages the worker to remain near the origin of the latent space, so that it can quickly achieve any goal by moving a small amount in the

| Method | Task | Observations | | Training | Evaluation |
|--------|------|--------------|--|----------|------------|
| HIRO |  |  | • Proprioception
• Goal XY pos
• Robot XY pos | Random XY goals, dense L2 reward | Fixed XY goal, success rate |
| NORL |  |  | • Proprioception
• Goal XY pos
• Top view (5×5) | Random XY goals, dense L2 reward | Fixed XY goal, success rate |
| This paper |  |  | • Proprioception
• First-person camera (64×64×3) | End-to-end reinforcement learning, sparse reward when the robot is touching the goal | |

**Figure 3:** Comparison of Ant Mazes in the literature and this paper. HIRO (Nachum et al., 2018a) provided global XY coordinates of the goal and robot position to the agent and trained with dense rewards on uniformly sampled training goals. NORL (Nachum et al., 2018b) replaced the robot XY position with a global top-down view of the environments, downsampled to 5×5 pixels. In this paper, we tackle the more challenging problem of learning directly from an egocentric camera without global information provided to the agent, and only give a sparse reward for time steps where the robot is at the goal. To succeed at these tasks, an agent has to autonomously explore the environment and identify landmarks to localize itself and navigate the mazes.

right direction. We thus incorporate the idea that the magnitude of both vectors should be similar into the cosine similarity, resulting in our *max-cosine* reward:

$$r_t^{\text{goal}} \doteq (g/m)^T (s_{t+1}/m) \quad \text{where} \quad m \doteq \max(\|g\|, \|s_{t+1}\|) \tag{8}$$

When the state and goal vectors are of the same magnitude, the reward simplifies to the cosine similarity. When their magnitudes differ, the vectors are both normalized by the larger of the two magnitudes, and thus the worker receives a down-scaled cosine similarity as the reward. As a result, the worker is incentivized to match the angle and magnitude of the goal. Unlike the L2 similarity, the reward scale of our goal reward is not affected by the scale of the underlying feature space.

The worker maximizes only the goal rewards. We make this design choice to demonstrate that the interplay between the manager and the worker is successful across many environments, although we also include an ablation experiment where the worker additionally receives task reward, which further improves performance. The worker is optimized by Dreamer with a goal-conditioned state-value critic. For updating the worker, we cut the imagined rollouts into distinct trajectories of length $K$ within which the goal is constant. The state-critic estimates goal rewards beyond this horizon under the same goal, allowing the worker to learn to reach far-away goals.

## 3 Experiments

We evaluate Director on two challenging benchmark suites with visual inputs and very sparse rewards, which we expect to be challenging to solve using a flat policy without hierarchy (Section 3.1). We further evaluate Director on a wide range of standard tasks from the literature to demonstrate its generality and ensure that the hierarchy is not harmful in simple settings (Section 3.2). We use a fixed set of hyperparameters not only across tasks but also across domains, detailed in Table F.1. Finally, we offer insights into the learned hierarchical behaviors by visualizing the latent goals selected during environment episodes (Section 3.3). Ablations and additional results are included in the appendix.

**Implementation**  We implemented Director on top of the public source code of DreamerV2 (Hafner et al., 2020a), reusing its default hyperparameters. We additionally increased the number of environment instances to 4 and set the training frequency to one gradient step per 16 policy steps, which drastically reduced wall-clock time and decreased sample-efficiency mildly. Implementing Director in the code base amounts to about 250 lines of code. The computation time of Director is 20% longer than that of DreamerV2. Each training run used a single V100 GPU with XLA and mixed precision enabled and completed in less than 24 hours. All our agents and environments will be open sourced upon publication to facilitate future research in hierarchical reinforcement learning.

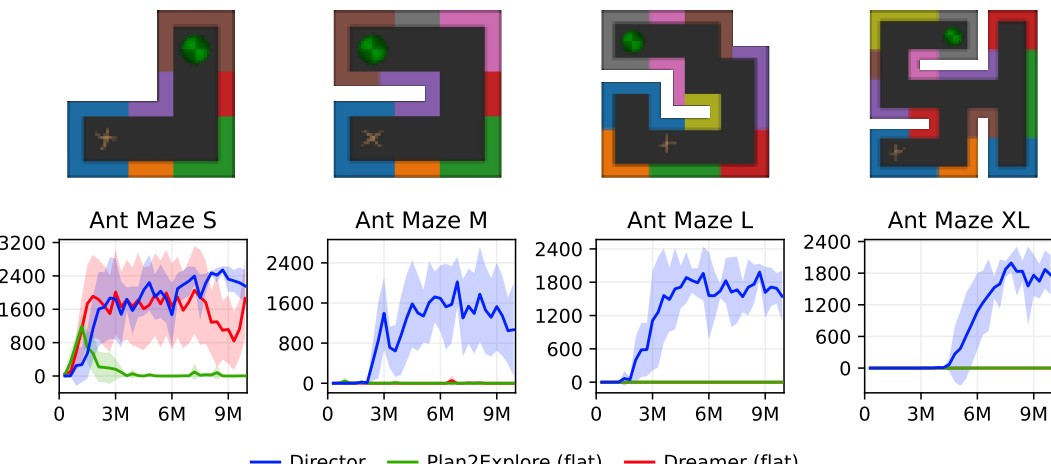

**Figure 4:** Egocentric Ant Maze benchmark. A quadruped robot is controlled through joint torques to navigate to a fixed location in a 3D maze, given only first-person camera and proprioceptive inputs. This is in contrast to prior benchmarks where the agents received their global XY coordinate or top-down view. The only reward is given at time steps where the agent touches the reward object. Plan2Explore fails in the small maze because the robot flips over too much, a common limitation of low-level exploration methods. Director solves all four tasks by breaking them down into manageable subgoals that the worker can reach, while learning in the end-to-end reinforcement learning setting.

**Baselines** To fairly compare Director to the performance of non-hierarchical methods, we compare to the DreamerV2 agent on all tasks. DreamerV2 has demonstrated strong performance on Atari (Hafner et al., 2020a), Crafter (Hafner, 2021) and continuous control tasks (Yarats et al., 2021) and outperforms top model-free methods in these domains. We kept the default hyperparameters that the authors tuned for DreamerV2 and did not change them for Director. In addition to its hierarchical policy, Director employs an exploration bonus at the top-level. To isolate the effects of hierarchy and exploration, we compare to Plan2Explore (Sekar et al., 2020), which uses ensemble disagreement of forward models as a directed exploration signal. We combined the extrinsic and exploration returns after normalizing by their exponential moving standard deviation with weights 1.0 and 0.1, as in Director. We found Plan2Explore to be effective across both continuous and discrete control tasks.

### 3.1 Sparse Reward Benchmarks

**Egocentric Ant Mazes** Learning navigation tasks directly from joint-level control has been a long-standing milestone for reinforcement learning with sparse rewards, commonly studied with quadruped robots in maze environments (Florensa et al., 2017; Nachum et al., 2018a). However, previous attempts typically required domain-specific inductive biases to solve such tasks, such as providing global XY coordinates to the agent, easier practice goals, and a ground-truth distance reward, as summarized in Figure 3. In this paper, we instead attempt learning directly from first-person camera inputs, without privileged information, and a single sparse reward that the agent receives while in the fixed goal zone. The control frequency is 50Hz and episodes end after a time limit of 3000 steps. There are no early terminations that could leak task-information to the agent (Laskin et al., 2022). To help the agent localize itself from first-person observations, we assign different colors to different walls of the maze.

As shown in Figure 4, we evaluate the agents in four mazes that span varying levels of difficulty. Because of the sparse reward, the episode returns correspond to the number of time steps for which the agent remains at the goal after reaching it. Curves show the mean and standard deviation across 5 independent seeds. We find that the flat Dreamer agent succeeds at the smallest of the four mazes, and the flat exploration policy Plan2Explore makes some initial learning progress but fails to converge to the optimal solution. Inspecting the trajectories revealed that Plan2Explore chooses too chaotic actions that often result in the robot flipping over. None of the baselines learn successful behaviors in the larger mazes, demonstrating that the benchmark pushes the limits of current approaches. In contrast, Director discovers and learns to reliably reach the goal zone at all four difficulty levels, with the larger maze taking longer to master.

**Visual Pin Pads** The Pin Pad suite of environments is designed to evaluate an agent's ability to explore and assign credit over long horizons, isolated from the complexity of 3D observations or

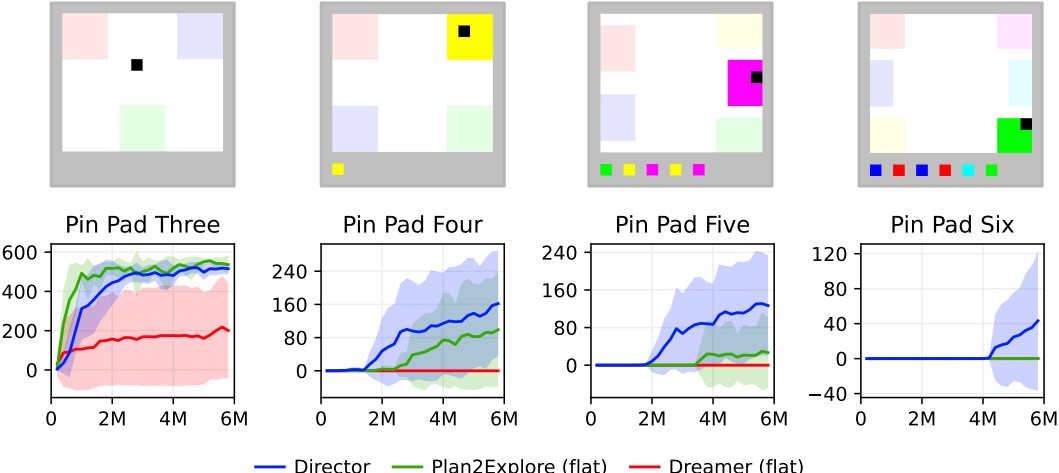

**Figure 5:** Visual Pin Pad benchmark. The agent controls the black square to move in four directions. Each environment has a different number of pads that can be activated by walking to and stepping on them. A single sparse reward is given when the agent activates all pads in the correct sequence. The history of previously activated pads is shown at the bottom of the screen. Plan2Explore uses low-level exploration and performs well in this environment, but struggles for five and six pads, which requires more abstract exploration and longer credit assignment. Director learns successfully across all these environments, demonstrating its benefit on this long-horizon benchmark.

sophisticated joint-level control. As shown in Figure 5, each environment features a moving black square that the agent can control in all four directions and fixed pads of different colors that agent can activate by walking over to the pad and stepping on it. The task requires discovering the correct sequence of activating all pads, at which point the agent receives a sparse reward of 10 points and the agent position is randomly reset. Episodes last for 2000 steps and there are no intermediate rewards for activating the pads. To remove the orthogonal challenge of learning long-term memory (Gregor et al., 2019), the history of previously activated pads is displayed at the bottom of the image.

The agent performance is shown in Figure 5, which displays mean and standard deviation across five independent seeds. The easiest environment contains three pads, so the agent only has to decide whether to activate the pads in clockwise or counter-clockwise sequence. The flat Dreamer agent sometimes discovers the correct sequence. The exploration bonus of Plan2Explore offers a significant improvement over Dreamer on this task. Dreamer fails to discover the correct sequence in the harder environments that contain more pads. Plan2Explore still achieves some reward with four pads, struggles with five pads, and completely fails with six pads. In contrast, Director discovers the correct sequence in all four environments, demonstrating the benefit of hierarchy over flat exploration.

### 3.2 Standard Benchmarks

To evaluate the robustness of Director, we train on a wide range of standard benchmarks, which typically require no long-term reasoning. We choose Atari games (Bellemare et al., 2013), the Control Suite from pixels (Tassa et al., 2018), Crafter (Hafner, 2021), and tasks from DMLab (Beattie et al., 2016) to cover a spectrum of challenges, including continuous and discrete actions and 2D and 3D environments. We compare two versions of Director. In its standard variant, the worker learns purely from goal rewards. This tests the ability of the manager to propose successful goals and the ability of the worker to achieve them. In the second variant, the worker learns from goal and task returns with weights $1.0$ and $0.5$, allowing the worker to fill in low-level details in a task-specific manner, which the manager may be too coarse-grained to provide. Ideally, we would like to see Director not perform worse than Dreamer on any task when giving task reward to the worker.

The results of this experiment are summarized in Appendix A due to space constraints, with the full training curves for Atari and the Control Suite included in Appendices J and K. We observe that Director indeed learns successfully across many environments, showing broader applicability than most prior hierarchical reinforcement learning methods. In addition, providing task reward to the worker is not as important as expected — the hierarchy solves a wide range of tasks purely by following goals at the low level. Additionally providing task reward to the worker completely

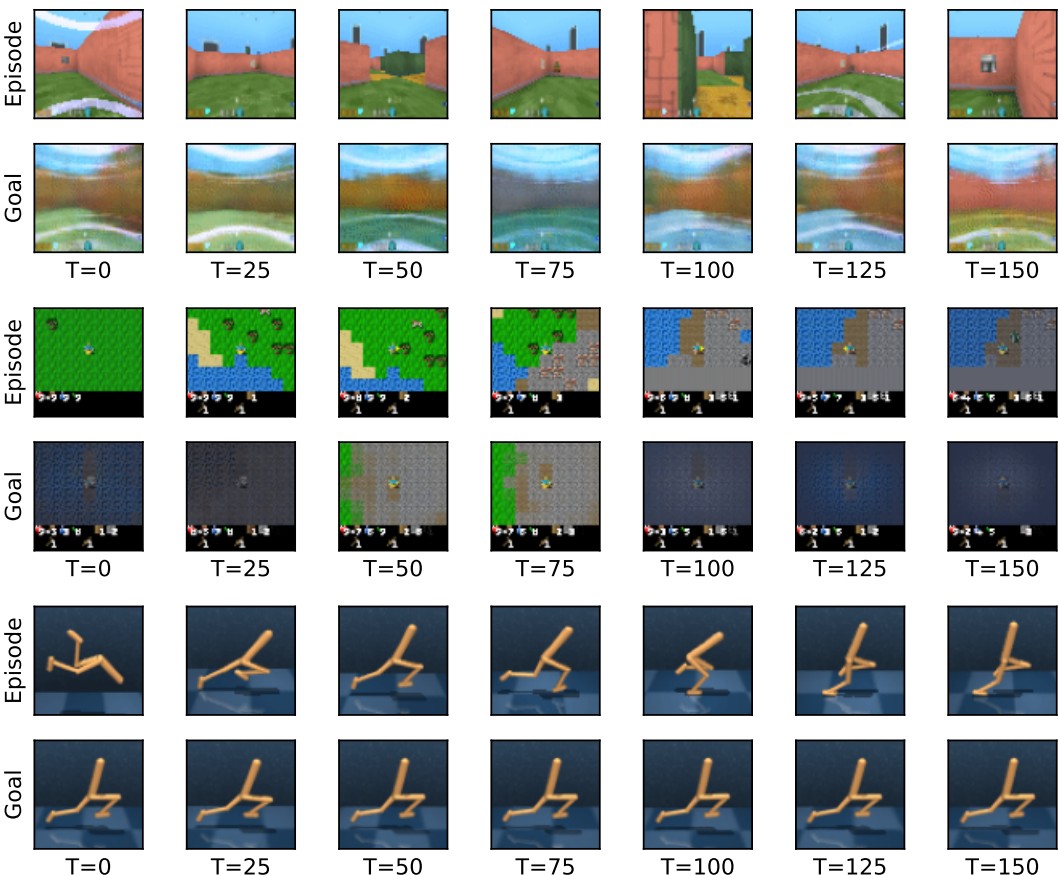

**Figure 6:** Subgoals discovered by Director. For interpretability, we decode the latent goals into images using the world model. **Top:** In DMLab, the manager chooses goals of the teleport animation that occurs every time the agent has collected the reward object, which can be seen in the episode at time step 125. **Middle:** In Crafter, the manager first directs the worker via the inventory display to collect wood and craft a pickaxe and a sword, with the worker following command. The worker then suggests a cave via the terrain image to help the worker find stone. As night breaks, it suggests hiding from the monsters in a cave or on a small island. **Bottom:** In Walker, the manager abstracts away the detailed leg movements by suggesting a forward leaning pose and a shifting floor pattern, with the worker successfully filling in the joint movements. Fine-grained subgoals are not required for this task, because the horizon needed for walking is short enough for the worker. Videos are available on the project website: https://danijar.com/director/

closes the gap to the state-of-the-art DreamerV2 agent. Figure K.1 in the appendix further shows that Director achieves a higher human-normalized median score than Dreamer on the 55 Atari games.

## 3.3 Goal Interpretations

To gain insights into the decision making of Director, we visualize the sequences of goals it selects during environment interaction. While the goals are latent vectors, the world model allows us to decode them into images for human inspection. Figures 1 and 6 show episodes with the environment frames (what the agent sees) at the top and the decoded subgoals (what the manager wants the worker to achieve) at the bottom. Visualizations for additional environments are included in Appendix L.

- **Ant Maze M**  The manager chooses goals that direct the agent through the different sections of the maze until the reward object is reached. We also observed that initially, the manager chooses more intermediate subgoals whereas later during training, the worker learns to achieve further away goals and thus the manager can select fewer intermediate goals.
- **DMLab Goals Small**  The manager directs the worker to the teleport animation, which occurs every time the agent collects the reward object and gets teleported to a random location in the 3D maze. Time step $T = 125$ shows an example of the worker reaching the animation in the environment. Unlike the Ant Maze benchmark, navigating to the goal in DMLab requires no joint-level control and is simple enough for the worker to achieve without fine-grained subgoals.
- **Crafter**  The manager requests higher inventory counts of wood and stone materials, as well as wooden tools, via the inventory display at the bottom of the screen. It also directs the worker to a cave to find stone and coal but generally spends less effort on suggesting what the world around the agent should look like. As night breaks, the manager tasks the worker with finding a small cave or island to hide from the monsters that are about to spawn.
- **Walker Walk**  The manager abstracts away the detail of leg movement, steering the worker using a forward leaning pose with both feet off the ground and a shifting floor pattern. While the images cannot show this, the underlying goal vectors are Markovian states that can contain velocities, so it is likely that the manager additionally requests high forward velocity. The worker fills in the details of standing up and moving the legs to pass through the sequence of goals.

Across domains, Director chooses semantically meaningful goals that are appropriate for the task, despite using the same hyperparameters across all tasks and receiving no domain-specific knowledge.

## 4 Related Work

Approaches to hierarchical reinforcement learning include learning low-level policies on a collection of easier pre-training tasks (Heess et al., 2016; Tessler et al., 2017; Frans et al., 2017; Rao et al., 2021; Veeriah et al., 2021), discovering latent skills via mutual-information maximization (Gregor et al., 2016; Florensa et al., 2017; Hausman et al., 2018; Eysenbach et al., 2018; Achiam et al., 2018; Merel et al., 2018; Laskin et al., 2022; Sharma et al., 2019; Xie et al., 2020; Hafner et al., 2020b; Strouse et al., 2021), or training the low-level as a goal-conditioned policy (Andrychowicz et al., 2017; Levy et al., 2017; Nachum et al., 2018a;b; Co-Reyes et al., 2018; Warde-Farley et al., 2018; Nair et al., 2018; Pong et al., 2019; Hartikainen et al., 2019; Gehring et al., 2021; Shah et al., 2021). These approaches are described in more detail in Appendix I.

Relatively few works have demonstrated successful learning of hierarchical behaviors directly from pixels without domain-specific knowledge, such as global XY positions, manually specified pre-training tasks, or precollected diverse experience datasets. HSD-3 (Gehring et al., 2021) showed transfer benefits for low-dimensional control tasks. HAC (Levy et al., 2017) learned interpretable hierarchies but required semantic goal spaces. FuN (Vezhnevets et al., 2017) learned a two-level policy where both levels maximize task reward and the lower level is regularized by a goal reward but did not demonstrate clear benefits over an LSTM baseline (Hochreiter and Schmidhuber, 1997). We leverage explicit representation learning and temporally-abstract exploration, demonstrate substantial benefits over flat policies on sparse reward tasks, and underline the generality of our method by showing successful learning without giving task reward to the worker across many domains.

## 5 Discussion

We present Director, a reinforcement learning agent that learns hierarchical behaviors from pixels by planning in the latent space of a learned world model. To simplify the control problem for the manager, we compress goal representations into compact discrete codes. Our experiments demonstrate the effectiveness of Director on two benchmark suites with very sparse rewards from pixels. We also show that Director learns successfully across a wide range of different domains without giving task

reward to the worker, underlining the generality of the approach. Decoding the latent goals into images using the world model makes the decision making of Director more interpretable and we observe it learning a diverse range of strategies for breaking tasks down into subgoals.

**Limitations and future work**   We designed Director to be as simple as possible, at the expense of design choices that could restrict its performance. The manager treats its action space as a black box, without any knowledge that its actions correspond to states. For example, one could imagine regularizing the manager to choose goals that have a high value under its learned critic. Lifting the assumption of changing goals every fixed number of time steps, for example by switching based on a separate classifier or as soon as the previous goal has been reached, could enable the hierarchy to adapt to the environment and perform better on tasks that require precise timing. Moreover, goals are points in latent space, whereas distributional goals or masks would allow the manager to only specify targets for the parts of the state that are currently relevant. Learning temporally-abstract dynamics would allow efficiently learning hierarchies of more than two levels without having to use exponentially longer batches. We suspect that these ideas will improve the long-term reasoning of the agent. Besides improving the capabilities of the agent, we see disentangling the reasons for why Director works well, beyond the ablation experiments in the appendix, as promising future work.

**Acknowledgements**   We thank Volodymyr Mnih, Michael Laskin, Alejandro Escontrela, Nick Rhinehart, and Hao Liu for insightful discussions. We thank Kevin Murphy, Ademi Adeniji, Olivia Watkins, Paula Gradu, and Younggyo Seo for feedback on the initial draft of the paper.

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

## Societal Impact

Developing hierarchical reinforcement learning methods that are useful for real-world applications will still require further research. However, in the longer term future, it has the potential to help humans automate more tasks by reducing the need to specify intermediate rewards. Learning autonomously from sparse rewards has the benefit of reduced human effort and less potential for reward hacking, but reward hacking is nonetheless a possibility that will need to be addressed once hierarchical reinforcement learning systems become more powerful and deployed around humans.

As reinforcement learning techniques become deployed in real-world environments, they will be capable of causing direct harm, intentional or unintentional. Director does not attempt to directly address such safety issues, but the ability to decode its latent goals into images for human inspection provides some transparency to the decisions the agent is making. This transparency may permit auditing of failures after-the-fact, and may additionally support interventions in some situations, if a human or automated observer is in a position to monitor the high-level goals Director generates.

