# A  Standard Benchmarks

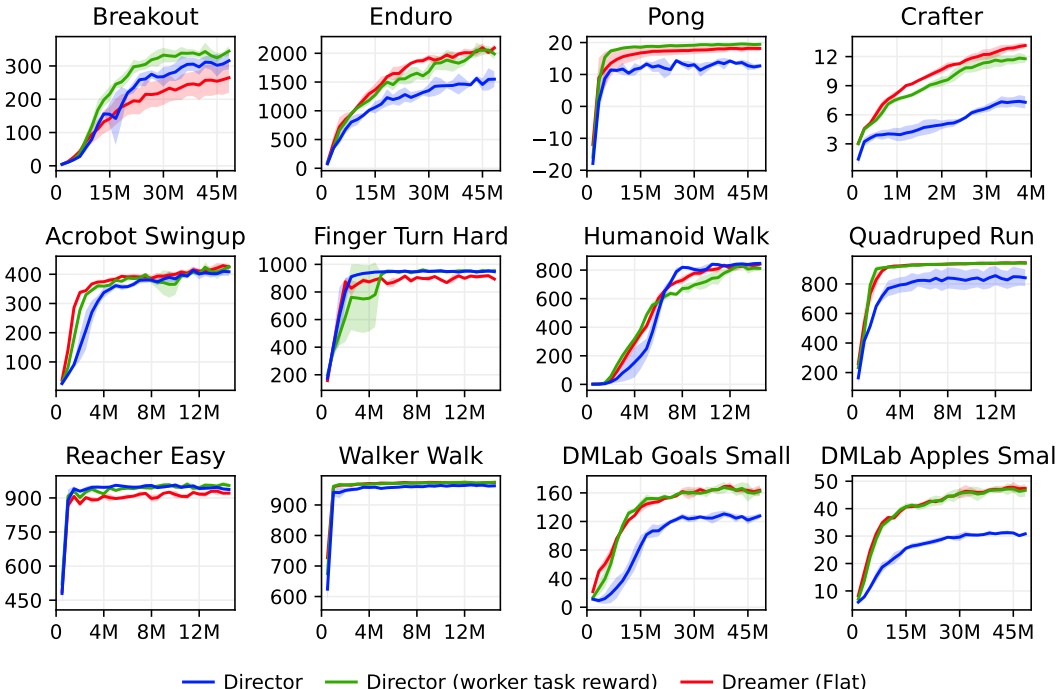

**Figure A.1:** Evaluation of Director on standard benchmarks, showing that Director is not limited to sparse reward tasks but is generally applicable. Director learns successfully across Atari, Crafter, Control Suite, and DMLab. This is an accomplishment because the worker receives no task reward and is purely steered through sub-goals selected by the manager. When additionally providing task reward to the worker, performance matches that of the state-of-the-art model-based agent Dreamer.

# B  Ablation: Goal Autoencoder

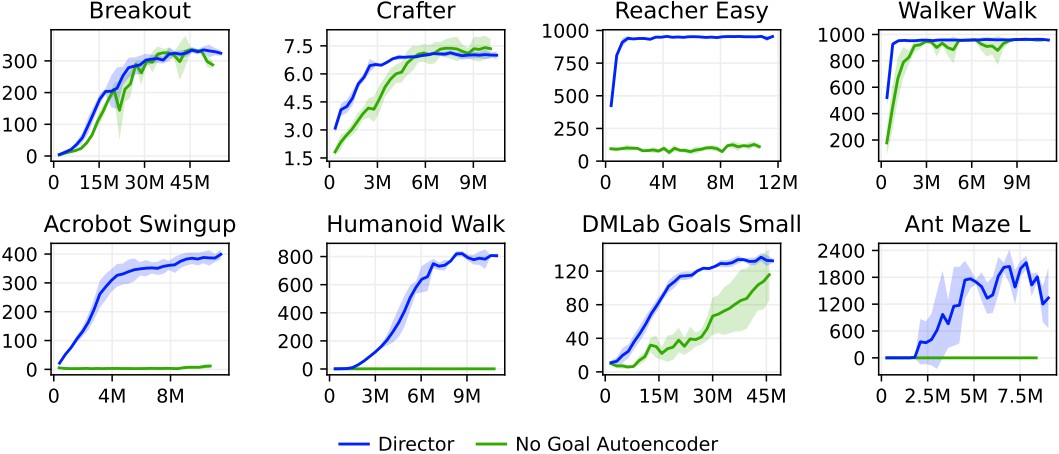

**Figure B.1:** Ablation of the Goal Autoencoder used by Director (Section 2.2). We compare the performance of Director to that of a hierarchical agent where the manager directly chooses 1024-dimensional goals in the continuous representation space of the world model. We observe that this simplified approach works surprisingly well in some environments but fails at environments with sparser rewards, likely because the control problem becomes too challenging for the manager.

# C   Ablation: Goal Reward

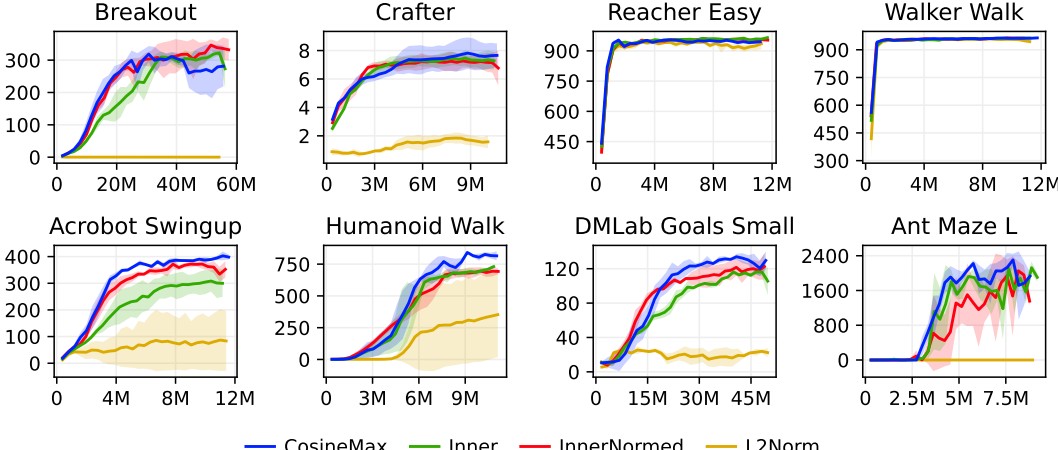

**Figure C.1:** Ablation of the goal reward used by Director. We compare the performance of Director with cosine-max reward (Equation 8) to alternative goal similarity functions. *Inner* refers to the inner product $g^T s_{t+1}$ without any normalization or clipping, which results in different reward scale based on the goal magnitude and encourages the worker to overshoot its goals in magnitude. *InnerNormed* refers to $(g/\|g\|)^T(s_{t+1}/\|g\|)$ where the goal and state are normalized by the goal magnitude, which normalizes the reward scale across goals but still encourages the worker to overshoot its goals. *L2Norm* is the negative euclidean distance $-\|g - s_{t+1}\|$. We observe that Director is robust to the precise goal reward, with the three rewards based on inner products performing well across tasks. The L2 reward works substantially worse. We hypothesize the reason to be that inner products allow goals to ignore some state dimensions by setting them to zero, whereas setting dimensions to zero for the L2 reward still requires the worker to care about them.

# D   Ablation: Exploration

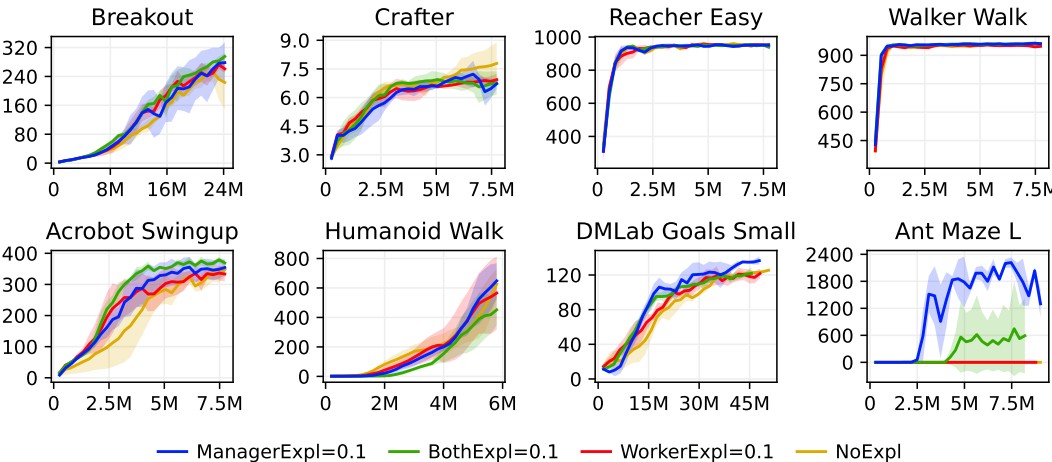

**Figure D.1:** Ablation of where to apply the exploration bonus in Director (Equation 6). We compare the performance of Director with exploration reward for only the manager, for only the worker, for both, and for neither of them. We find giving the exploration bonus to the manager, which results in temporally-abstract exploration, is required for successful learning in the Ant Maze, and that additional low-level exploration hurts because it results in too chaotic leg movements. In standard benchmarks that typically require short horizons, the exploration bonus is not needed.

# E Pseudocode

---
**Algorithm 1:** Director
---

1  Initialize replay buffer and neural networks.
2  **while** *not converged* **do**

      `// Acting`
3      Update model state $s_t \sim \mathrm{repr}(s_t \mid s_{t-1}, a_{t-1}, x_t)$.

4      **if** $t \bmod K = 0$ **then**
            `// Update internal goal`
5            Sample abstract action $z \sim \mathrm{mgr}(z \mid s_t)$.
6            Decode into model state $g = \mathrm{dec}(z)$.

7      Sample action $a_t \sim \mathrm{wkr}(a_t \mid s_t, g)$.
8      Send action to environment and observe $r_t$ and $x_{t+1}$.
9      Add transition $(x_t, a_t, r_t)$ to replay buffer.

      `// Learning`
10     **if** $t \bmod E = 0$ **then**

            `// World Model`
11           Draw sequence batch $\{(x, a, r)\}$ from replay buffer.
12           Update world model on batch (Equation 2) and get states $\{s\}$.

            `// Goal Autoencoder`
13           Update goal autoencoder on $\{s\}$ (Equation 4).

            `// Policies`
14           Imagine trajectory $\{(\hat{s}, \hat{a}, \hat{g}, \hat{z})\}$ under model and policies starting from $\{s\}$.
15           Predict extrinsic rewards $\{\mathrm{rew}(s)\}$.
16           Compute exploration rewards $\{r^{\mathrm{expl}}\}$ (Equation 6).
17           Compute goal rewards $\{r^{\mathrm{goal}}\}$ (Equation 8).
18           Abstract trajectory to update manager (Equations 11 and 12).
19           Split trajectory to update worker (Equations 11 and 12).

---

# F Hyperparameters

| Name | Symbol | Value |
|------|--------|-------|
| Parallel envs | — | 4 |
| Training every | $E$ | 16 |
| MLP size | — | $4 \times 512$ |
| Activation | — | $\mathrm{LayerNorm} + \mathrm{ELU}$ |
| Imagination horizon | $H$ | 16 |
| Discount factor | $\gamma$ | 0.99 |
| Goal duration | $K$ | 8 |
| Goal autoencoder latents | $L$ | 8 |
| Goal autoencoder classes | $C$ | 8 |
| Goal autoencoder beta | $\beta$ | 1.0 |
| Learning rate | — | $10^{-4}$ |
| Weight decay | — | $10^{-2}$ |
| Adam epsilon | $\epsilon$ | $10^{-6}$ |

**Table F.1:** Hyperparameters of Director. We use the same hyperparameters across all experiments. The hyperparameters for training the world model and optimizing the policies were left unchanged compared to DreamerV2 (Hafner et al., 2020a).

# G  Vector of Categoricals

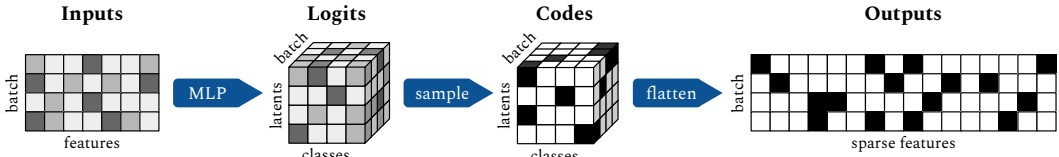

**Figure G.1:** Director uses a variational autoencoder to turn its state representations into discrete tokens that are easy to select between for the manager policy. For this, we use the vector of categoricals approach of DreamerV2 (Hafner et al., 2020a). For a given input vector, the encoder outputs a matrix of logits of $L$ latent dimensions with $C$ classes each. We sample from the logits and one-hot encode the result to obtain a sparse matrix of the same shape as the logits. To backpropagate gradients through this step, we simply use the gradient with respect to the one-hot matrix as the gradient with respect to the categorical probabilities (Bengio et al., 2013). The matrix is then flattened, resulting in a sparse representation with $L$ out of the $L \times C$ feature dimensions set to one.

# H  Policy Optimization

Both the worker and manager policies of Director are optimized using the actor critic algorithm of Dreamer (Hafner et al., 2019; 2020a). For this, we use the world model to imagine a trajectory in its compact latent space. The actor network is optimized via policy gradients (Williams, 1992) with a learned state-value critic for variance reduction and to estimate rewards beyond the rollout:

$$\text{Actor:} \quad \pi(a_t \mid s_t) \qquad \text{Critic:} \quad v(s_t) \tag{9}$$

The world model allows cheaply generating as much on-policy experience as needed, so no importance weighting or clipping applies (Schulman et al., 2017). To train both actor and critic from an imagined trajectory of length $H$, $\lambda$-returns are computed from the sequence of rewards and predicted values (Sutton and Barto, 2018):

$$V_t^\lambda \doteq r_t + \gamma\Big((1 - \lambda)v(s_{t+1}) + \lambda V_{t+1}^\lambda\Big), \quad V_H^\lambda \doteq v(s_H). \tag{10}$$

The $\lambda$-returns are averages over multi-step returns of different lengths, thus finding a trade-off between incorporating further ahead rewards quickly and reducing the variance of long sampled returns. The critic is learned by regressing the $\lambda$-returns via a squared loss, where $\text{sg}(\cdot)$ indicates stopping the gradient around the targets:

$$\mathcal{L}(v) \doteq \mathrm{E}_{p_\phi, \pi}\Big[ \sum_{t=1}^{H-1} \tfrac{1}{2}\big(v(s_t) - \text{sg}(V_t^\lambda)\big)^2 \Big]. \tag{11}$$

The actor is updated by policy gradients on the same $\lambda$-returns, from which we subtract the state-value $v(s_t)$ as a baseline that does not depend on the current action for variance reduction. The second term in the actor objective, weighted by the scalar hyperparameter $\eta$, encourages policy entropy to avoid overconfidence and ensures that the actor explores different actions:

$$\mathcal{L}(\pi) \doteq -\mathrm{E}_{\pi, p_\phi}\Big[ \sum_{t=1}^{H} \ln \pi(a_t \mid s_t)\,\text{sg}(V_t^\lambda - v(s_t)) + \eta\,\mathrm{H}\big[\pi(a_t \mid s_t)\big] \Big] \tag{12}$$

When there are multiple reward signals, such as the task and exploration rewards for the manager, we learn separate critics (Burda et al., 2018) for them and compute separate returns, which we normalize by their exponential moving standard deviation with decay rate 0.999. The baselines are normalized by the same statistics and the weighted average of the advantages is used for updating the policy.

# I   Further Related Work

**Pretraining tasks**   One way to integrate domain knowledge into hierarchical agents is to learn primitives on simpler tasks and then compose them to solve more complex tasks afterwards. Learning primitives from manually specified tasks simplifies learning but requires human effort and limits the generality of the skills. DSN (Tessler et al., 2017) explicitly specifies reward functions for the low-level policies and then trains a high-level policy on top to solve Minecraft tasks. MLSH (Frans et al., 2017) pretrains a hierarchy with separate low-level policies on easier tasks, alternating update phases between the two levels. HeLMS (Rao et al., 2021) learns reusable robot manipulation skills from a given diverse dataset. MODAC (Veeriah et al., 2021) uses meta gradients to learn low-level policies that are helpful for solving tasks. However, all these approaches require manually specifying a diverse distribution of training tasks, and it is unclear whether generalization beyond the training tasks will be possible. Instead of relying on task rewards for learning skills, Director learns the worker as a goal-conditioned policy with a dense similarity function in feature space.

**Mutual information**   Mutual information approaches allow automatic discovery of skills that lead to future states. VIC (Gregor et al., 2016) introduced a scalable recipe for discovering skills by rewarding the worker policy for reaching states from which the latent skill can be accurately predicted, effectively clustering the trajectory space. Several variations of this approach have been developed with further improvements in stability and diversity of skills, including SSN4HRL (Florensa et al., 2017), DIAYN (Eysenbach et al., 2018) and VALOR (Achiam et al., 2018). DISCERN (Warde-Farley et al., 2018) and CIC (Laskin et al., 2022) learn a more flexible similarity function between skills and states through contrastive learning. DADS (Sharma et al., 2019) estimates the mutual information in state-space through a contrastive objective with a learned model. DISDAIN overcomes a collapse problem by incentivizing exploration through ensemble disagreement (Strouse et al., 2021). LSP (Xie et al., 2020) learns a world model to discover skills that have a high influence on future representations rather than inputs. While these approaches are promising, open challenges include learning more diverse skills without dropping modes and learning skills that are precise enough for solving tasks.

**Goal reaching**   Learning the low-level controller as a goal-conditioned policy offers a stable and general learning signal. HER (Andrychowicz et al., 2017) popularized learning goal-conditioned policies by combining a sparse reward signal with hindsight, which HAC (Levy et al., 2017) applied to learn an interpretable hierarchy with three levels by manually designing task-relevant goal spaces for each task. Instead of relying on hindsight with a sparse reward, HIRO (Nachum et al., 2018a) employs a dense reward in observation space based on the L2 norm and solves Ant Maze tasks given privileged information. NORL (Nachum et al., 2018b) introduces a representation learning objective to learn a goal space from a low-resolution top-down image instead but still uses a dense ground-truth reward for learning the manager. HSD-3 (Gehring et al., 2021) also learns a hierarchy with three levels and uses the robot joint space combined with a mask to specify partial goals. While not learning a high-level policy, RIG (Nair et al., 2018) computes goal rewards in the latent space of an autoencoder, allowing them to reach simple visual goals. FuN (Vezhnevets et al., 2017) provides task and goal rewards to its low-level policy and is trained from pixels but provides limited benefits over a flat policy trained on task reward. SecTAR (Co-Reyes et al., 2018) learns a sequence autoencoder that serves to propose goals and compute distances for low-dimensional environments and shows fast learning in low-dimensional environments with sparse rewards by high-level planning. DDL (Hartikainen et al., 2019) learns to reach goals from pixels by learning a temporal distance function. LEXA (Mendonca et al., 2021) learns a goal conditioned policy inside of a world model that achieves complex multi-object goal images, but assumes goals to be specified by the user. By comparison, Director solves difficult sparse reward tasks end-to-end from pixels without requiring domain-specific knowledge by learning a world model.

## J    Full Visual Control Suite

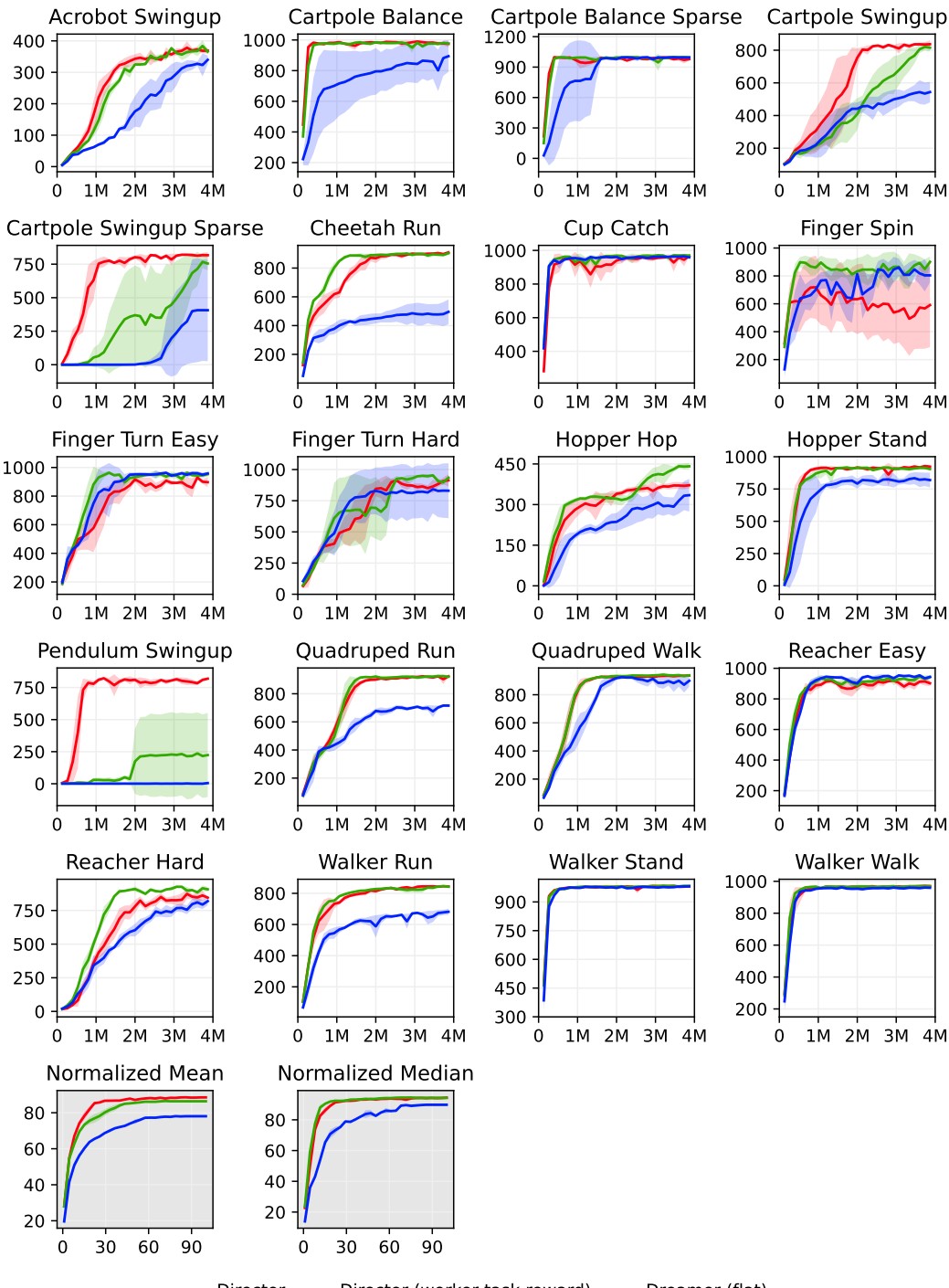

**Figure J.1:** To test the generality of Director, we evaluate it on a diverse set of 20 visual control tasks (Tassa et al., 2018) without action repeat. We find that Director solves a wide range of tasks despite giving no task reward to the worker, for the first time in the literature demonstrating a hierarchy with task-agnostic worker performing reliably across many tasks. When additionally providing task reward to the worker, performance reaches the state-of-the-art of Dreamer and even exceeds it on some tasks. These experiments use no action repeat and train every 16 environment steps, resulting in faster wall clock time and lower sample efficiency than the results reported by Hafner et al. (2019).

# K Full Atari Suite

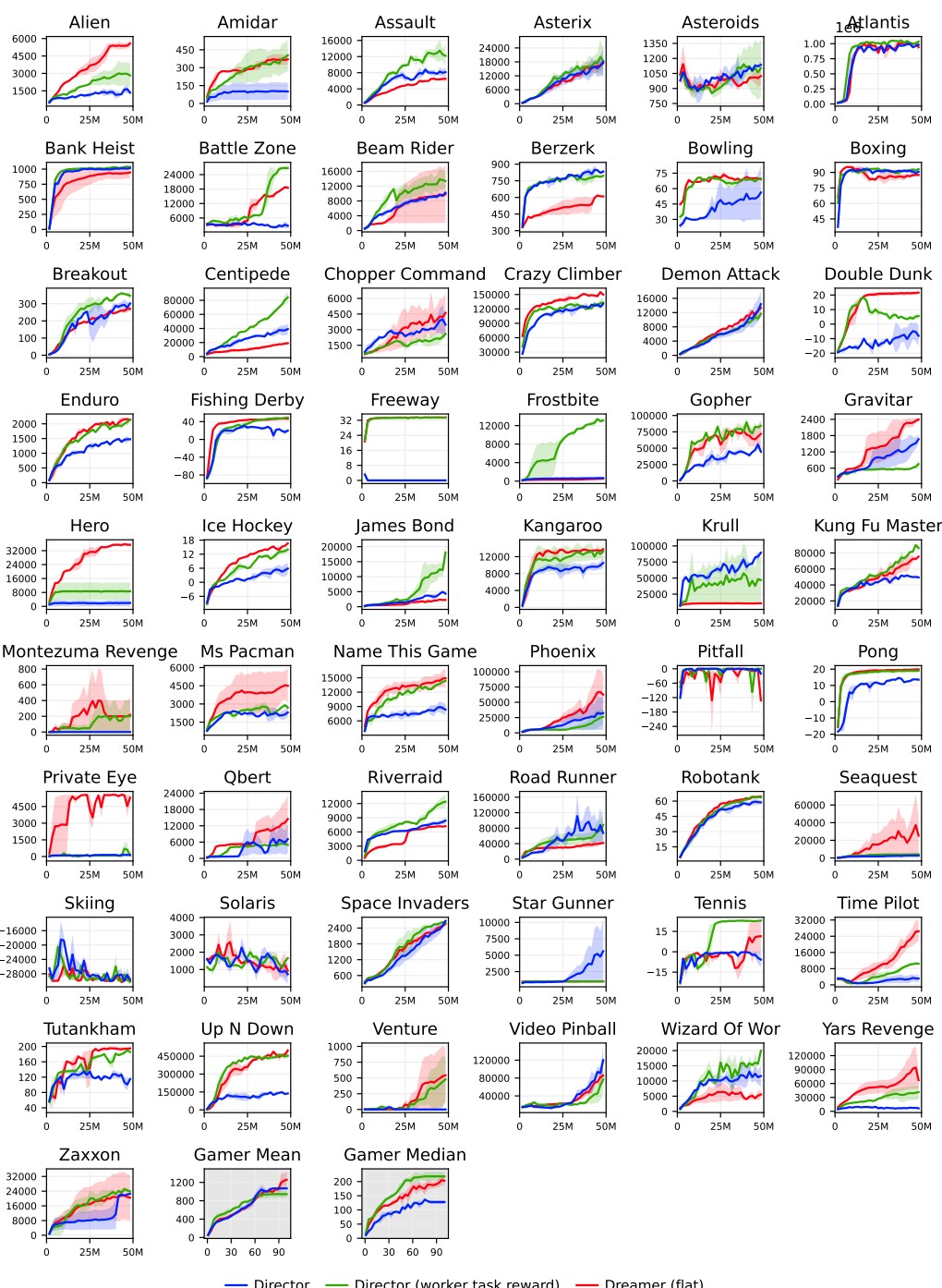

**Figure K.1:** To test the generality of Director, we evaluate it on 55 Atari games ([Bellemare et al., 2013](#)). We find that Director solves a wide range of tasks despite giving no task reward to the worker. Director is the first approach in the literature that demonstrates a hierarchy with task-agnostic worker performing reliably across many tasks. When additionally providing task reward to the worker, performance reaches that of Dreamer and even exceeds its human-normalized median score. These experiments use no action repeat and train every 16 environment steps, resulting in faster wall clock time and lower sample efficiency than the results reported by [Hafner et al. (2019)](#).

# L Additional Goal Visualizations

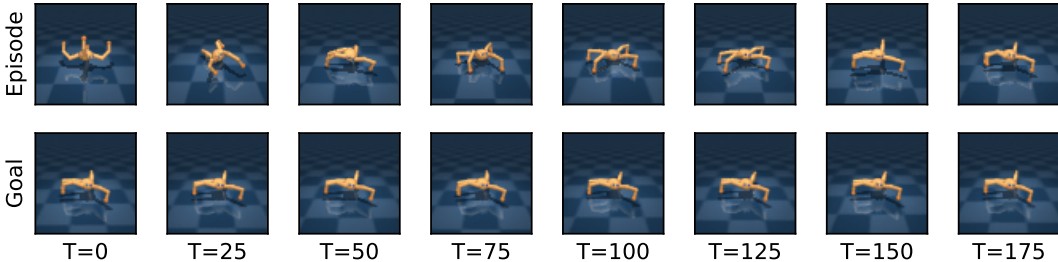

**(a)** Quadruped Walk. The manager learns to abstract away leg movements by requesting a forward leaning pose with shifting floor pattern, and the worker fills in the leg movements. The underlying latent goals are Markov states and thus likely contain a description of high forward velocity.

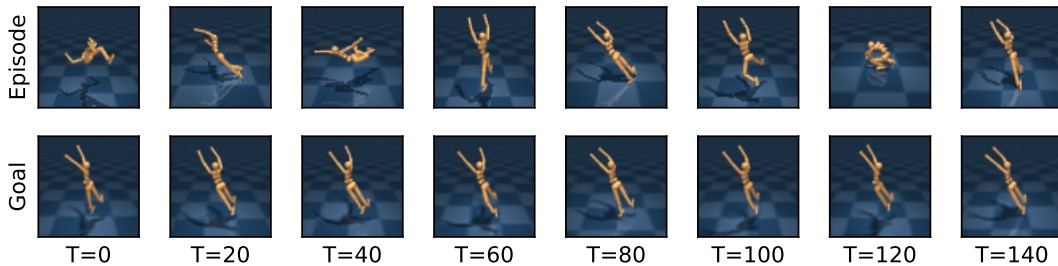

**(b)** Humanoid Walk. The manager learns to direct the worker via an extended pose with open arms, which causes the worker to perform energetic forward jumps that are effective at moving the robot forward. While the visualizations cannot show this, the underlying latent goals are Markovian and can contain velocity information.

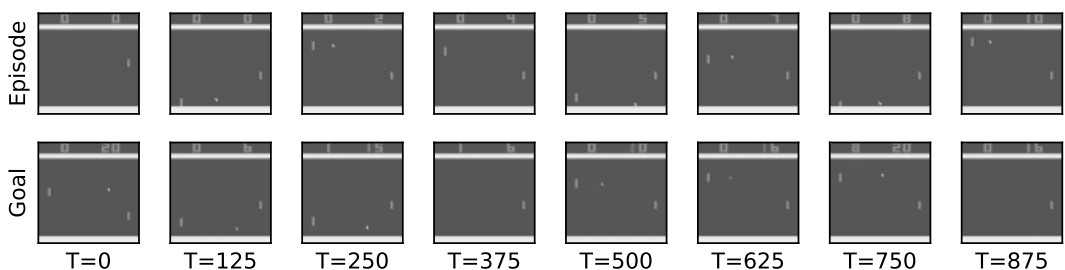

**(c)** Pong. The manager directs the worker simply by requesting a higher score via the score display at the top of the screen. The worker follows the command by outplaying the opponent. We find that for tasks that are easy to learn for the worker, the manager frequently chooses this hands-off approach to managing.

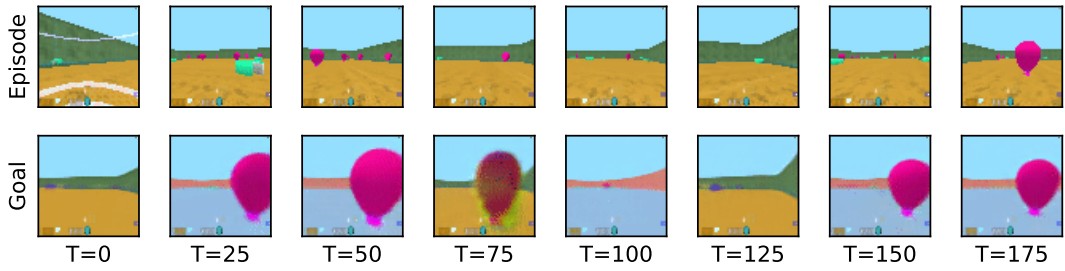

**(d)** DMLab Collect Good. The manager steers through the arena by targeting the good objects and avoiding the bad ones. The worker succeeds at reaching the balloons regardless of the wall textures, showing that it learns to focus on achievable aspects of the goals without being distracted by non-achievable aspects.

**Figure L.1:** Additional goal visualizations.