# OpenReview forum: "Deep Hierarchical Planning from Pixels"
_NeurIPS.cc/2022/Conference — NeurIPS 2022 Accept_

### Official Review · Reviewer_q27r · 2022-07-11

**Rating:** 6
**Confidence:** 4
**Soundness:** 3 good
**Presentation:** 4 excellent
**Contribution:** 2 fair

**Summary:**

Summary: Learn sub-policies by planning in the latent space of a learned dynamics model, where the low level policy performs goal-based RL, and the high level policy uses exploration and task rewards. It takes pixels as input, learns a latent world model, and then performs planning in the latent space of the world model. The world model is learned using a network dynamics modeler RSSM, from PlaNet, which is a variational pixel and rewards reconstruction algorithm. In order to reduce the dimensionality of the action space for the high-level controller, the algortihm uses a second encoder to encode into action space, which is the same as a vector of categoricals method used in Dreamer V2, which uses a variational reconstruction of the world model space. The exploration reward is based on reconstruction error (which mirrors work in curiosity rather than count-based methods), and combined with the extrinsic reward through  magic values. The worker policy uses a goal conditioned policy with a shaped max-cosine reward. The overall algorithm is evaluated on a large suite of common RL tasks including Atari, control suite, etc. against flat baselnes.

**Questions:**

What fundamental insights did implementing this give towards hierarchical RL?
How would semantic hierarchical methods compare against this as an upper bound, or does working from pixels provide an advantage.
What hindered the standard baselines from matching current flat methods?
How difficult was this to tune? What percentage of the development time was finding a good way to hook up the components?


**Limitations:**

This work does not describe the societal impact of an algorithm like this, though that description would be a philosophical exercise in whether better AI is a good thing. They also do not state clearly how the different components of Director could be interchanged, specifically the model-learning component, the model-based RL component, the exploration component and the low level goal reaching component, which would give more insight into the limitations and future work than describing a few edge cases.
I think the biggest limitations were in having a clear message of what the contribution of this work is, and that it is not clear whether this method is truly hierarchical, or just a complex way to learn flat policies with model-based RL.

**Strengths And Weaknesses:**

Strengths:
This work describes a straightforward concept well, utilizing prior work in model-based RL, goal-based RL and exploration to contruct a hierarchical RL algorithm. The stated goal of creating a simple algorithm is executed well, and evidenced by the ability to test the algorithm on a wide variety of different domains. On top of that, the design choices are generally very reasonable, with the model-based being used both for exploration and to generate a pre-latent space for the mutual-information like latent skills, and the low level policy being goal-following.
Considering the difficulty of running hierarchical RL algorithms, which incorporate the combination of a large number of different ideas functioning simultaniously, this work is an impressive achievement engineering, with finding the right combination of methods to get generally improved performance on a wide variety of tasks. The fact that the latent space can also be visualized because of the world model is a nice perk, and helps alleviate some questions related to whether the method actually utilizes an efficient division of labor.
Weaknesses:
The greatest disappointment with this paper is that it is more a feat of engineering than providing any significant insights into the way in which hierarchical RL can be run. Buried in the related work is the fact that feudal networks (Vezhnevets et. al. 2017), performed many of the same ideas with different design choices for generating the latent space and training the goal-based RL algorithm. While the design choices in this work appear to be superior, it is hard to tell directly from the experiments since there appear to be some gaps in terms of which tasks were tested on. As a result, the core insight of this paper appears to be that under these conditions we can combine feudal networks with dreamer-style model based RL.
This work utilizes a double-encoding which would have been interesting to investigate, at least in the appendix. While it makes sense to use a different embedding space for the world-model latent space (which needs all the information for reconstruction), and while the skill encoder finds a space that is a good size for learning, it would have been useful to get more information about some of the tradeoffs and design choices there. The ablation in the appendix only shows that it is necessary in some cases, not how the size of the goal autoencoder or other components scale.
Hyperparameters are always a significant question for hierarchical RL methods, and it isn't clear that all the (non-dreamer) hyperparameters are actually described in Appendix F, and more importantly, what the sensitivity is to those hyperparameters. It would also be nice to get the dreamer hyperparameters.
The Standard benchmark results (appendix A) is somewhat inconclusive. For one thing, 45M time steps is long compared to modern methods for atari games. As an example, Rainbow and PPO reach 400 reward (higher than the values given by Director and Dreamer), in 2M time steps (https://wandb.ai/tianshou/atari.benchmark/reports/Atari-Benchmark--VmlldzoxOTA1NzA5), which would equate to about 10x faster than the values given. Admittedly, Dreamer also has these issues, but as Breakout is the only atari task where Director has clear performance improvement over Dreamer, this puts into question performance in the whole domain.
Ant-maze domains often struggle with a sort of low-variance issue, which means that a policy that explores well enough to reach the goal eventually can reach the goal again. This is not usually an issue, but because the atari and other benchmark results are somewhat inconclusive, this brings up the question of whether the performance benefit is just because of exploration (dreamer never reaches the goal to send return back from because it fails to perform good exploration). The Plan2Explore results don't really alleviate this because they never reach the goal either, and comparison with a semantic baseline, or a count-based exploration baseline, would have been interesting. It does not need to outperform, just demonstrate that it did well.
The qualitative subgoal results are somewhat weak, because it isn't clear in the researcher is inferring meaning onto the subgoals, or the subgoals are actually chosen such that there is significant division of labor between the low and high level controllers. For the most part, it looks like the manager just chooses one goal, and all of the actual work is carried out by the low level agent. This hypothesis is further supported by the fact that when extrinsic reward is provided to the low level agent, performance often improves. This is not always the cas , but it appears to be the case often enough to call into question whether the work is actually utilizing hierarchy. Ironically, the PIN pad domain, the toy domain that is somewhat underdescribed, is probably the best place to show this, since having subgoals that lie on each of the PIN pads would pretty much demonstrate the result.
This work does not compare with other hierarchical baselines, in particular feudal networks which implements largely the same algorithm. Normally, this would be a huge problem, but historically hierarchical methods are a huge pain to implement, so it's somewhat understandable. However, at least an attempt to use feudal networks would have been insightful.

---

> ### Author Response · Authors · 2022-07-28
> **Author Response to Reviewer q27r (Part 2/2)**
>
> > more a feat of engineering than providing any significant insights into the way in which hierarchical RL can be run.
>
> The paper shows that goal-conditioned hierarchical RL can benefit greatly from compressing all the possible goals into a compact discrete space (goal autoencoder) and from employing an exploration bonus at the high level. Our corresponding ablations in Appendix B and D demonstrate the significant effect of these two ideas.
>
> We hypothesize that discretizing the goals simplifies learning for the high level by constraining it to choose among goals that correspond to valid states in the environment. While we see implementing these components into a widely successful hierarchical RL agent by itself as a substantial effort and contribution, we hypothesize that future hierarchical RL agents will likely benefit from the same two ideas.
>
> > feudal networks (Vezhnevets et. al. 2017), performed many of the same ideas with different design choices for generating the latent space and training the goal-based RL algorithm
>
> While the idea of using goal-conditioned policies for hierarchical RL has been around for a long time, there are many distinct differences between FeUdal Networks (FuN) and Director, in addition to the world model that you mentioned. Most important for the present discussion, FuN is not using any exploration bonus and FuN does not use a goal autoencoder to restrict its goals to realistic states. Both components are critical for the success of Director and our ablations that remove these components are closer to FuN. FuN also gives task rewards for the worker, so it is unclear whether the communication between the manager and the worker works reliably.
>
> We would have also liked to compare to FuN experimentally but unfortunately the authors did not release their code and previous attempts at a reimplementation at our lab were not promising. Combined with the results in the FuN paper showing only small improvements over an LSTM agent and that we are not aware of any follow-ups to FuN from DeepMind itself, we decided not to pursue this direction further.
>
> > 45M time steps is long compared to modern methods for atari games [...] Rainbow and PPO reach 400 reward (higher than the values given by Director and Dreamer), in 2M time steps (https://wandb.ai/tianshou [...]
>
> The aim of Appendix A is not to show benefits of Director over Dreamer. Instead, the point is that Director --- a method designed specifically for RL tasks with very sparse rewards --- can achieve competitive performance out of the box also on a wide range of tasks with dense rewards where the hierarchy isn't needed. Together with the sparse reward tasks in the main paper text, this shows that Director extends the set of environments solved by Dreamer, especially when giving some task reward to the worker.
>
> While it is possible to achieve faster learning on some tasks by increasing the rate of gradient steps or tuning specifically for easier tasks, this is orthogonal to the investigations in our paper. As mentioned in the experiments section, we fix the training frequency for all methods to perform a gradient step every 16 environment steps to allow for a fair comparison and fast experimentation.
>
> > Ant-maze domains often struggle with a sort of low-variance issue, which means that a policy that explores well enough to reach the goal eventually can reach the goal again. This is not usually an issue [...] this brings up the question of whether the performance benefit is just because of exploration [...] The Plan2Explore results don't really alleviate this because they never reach the goal either
>
> We don't fully understand this point, please clarify if our response doesn't resolve your question around this. We agree that the main challenge in environments with very sparse rewards is often exploration. In the larger Ant Mazes, 10M random actions receive not a single reward so some exploration objective is needed. In Ant Maze S, Plan2Explore reaches the goal a few times earlier on (see the non-zero return) but then becomes too explorative in its low-level actions and starts flipping over. In Pin Pad 3/4/5, Plan2Explore also clearly finds rewards (even when Dreamer does not). Sekar et al. (2019) showed that Plan2Explore outperforms ICM.

---

> ### Author Response · Authors · 2022-07-28
> **Author Response to Reviewer q27r (Part 1/2)**
>
> Thank you for your detailed review! In summary, we added our findings of investigating different goal autoencoders, point out the goal visualization videos on our anonymous website (linked in the abstract), and clarified on your questions. Please let us know if our response resolves your concerns or whether there are remaining points, in which case we would be happy to address them.
>
> > The qualitative subgoal results are somewhat weak [...] For the most part, it looks like the manager just chooses one goal, and all of the actual work is carried out by the low level agent.
>
> What you are describing is the fallback behavior of Director for dense reward tasks that don't require long horizons and hierarchy. In those cases, the worker alone can solve the task and thus the manager only has to communicate what the task is. On tasks that require long horizons (Visual Pin Pad, Egocentric Ant Maze), the subgoals correspond to intermediate steps along the task --- such as walls of different colors or different pads to step on --- that substantially simplify the task for the worker and enable Director to find rewards and repeatedly seek them out again in these very sparse tasks where other baselines fail.
>
> > the PIN pad domain [...] is probably the best place to show this, since having subgoals that lie on each of the PIN pads would pretty much demonstrate the result
>
> We agree that Visual Pin Pad is a good environment to show this. If you haven't yet, please take a look at the video visualizations of subgoals along the episode (on the anonymous website linked in the abstract). The high-level policy indeed chooses subgoals that correspond to which pad to activate next by adding the pad to the history display at the bottom of the screen. If you have further questions about the subgoal decomposition, please let us know and we'll be happy to go into more detail. The videos on the website also show meaningful subgoal decompositions on many other tasks.
>
> > the skill encoder [...] more information about some of the tradeoffs and design choices there. The ablation in the appendix only shows that it is necessary in some cases, not how the size of the goal autoencoder or other components scale.
>
> Thank you for this suggestion. We experimented with different latent spaces and we will include these findings in the final version of the paper:
>
> - Gaussian latents of different dimensions (32, 128). We found that the resulting unbounded latent space fails to constrain the manager to goals that correspond to valid states, similar to having no goal autoencoder at all.
> - A single categorical with a varying number of classes (8, 32, 128, 512). We found that single categoricals have not enough capacity to reasonably model the state space and result in poor performance except in very simple environments. Very large categoricals result in better modeling performance but create a large exploration challenge for the high-level policy, which also results in poor performance.
> - Vectors of categoricals of different sizes (4x4, 8x8, 16x16, 32x32). These factorized spaces have high capacity (8^8=16M) and also allow the high-level policy to explore multiple dimensions in parallel, simplifying its exploration problem. The different sizes all worked quite well with 8x8 performing best, which we used in the final agent.
>
> (continued below...)

---

> ### Author Response · Authors · 2022-08-07
> **Discussion**
>
> Dear Reviewer q27r,
>
> The discussion period is coming to an end soon and we haven't received a response from you yet. Could we please ask you to confirm whether our response has resolved your concerns or whether you see any remaining issues that motivate your current rating? If there are remaining issues, we would be more than happy to address them or further clarify where necessary.
>
> Thank you!

---

### Official Review · Reviewer_BArk · 2022-07-11

**Rating:** 6
**Confidence:** 4
**Soundness:** 3 good
**Presentation:** 3 good
**Contribution:** 3 good

**Summary:**

This paper proposed a method for learning a hierarchical policy that operates from pixels, without pre-defined high level actions, and using model based RL

The policy is broken up into 4 key components. A world model that models environment dynamics, representation, and reward. A manger policy that selects goals in a discrete latent space. A goal autoencoder that decodes the select goal into representation space, and a worker policy that selects low level actions to take to achieve the manager's goal.

The proposed method is evaluated in two environments that stress sparsity of rewards and on standard benchmarks.

In the sparse reward benchmarks, the proposed method outperforms baselines. The gap to baselines increases as the environment complexity increases.

On the standard benchmarks, the proposed method matches Dreamer when the worker is trained with the task-specific reward in addition to the reward that encourages it to reach the manager's goal.

**Questions:**

See above

**Limitations:**

Yes

**Strengths And Weaknesses:**

### Strengths

The proposed method is able to learn a hierarchical policy with MBRL that does not require pre-specified high level actions. These high-level actions are simply environment states to reach, making them easily interpretable.

The proposed method matches state-of-the-art on standard benchmarks.

The proposed method works well with sparse rewards.

There are extensive ablations in the supplement.

The paper is well-written.

### Weaknesses

The evaluation falls short of the initial promise in the intro. To the knowledge of the reviewer, none of the tasks studied have the length of time horizon promised in the intro. Thus it remains unclear if this method indeed works better in that promised setting. The proposed method is clearly helpful in settings with sparse rewards, so perhaps that should be the motivation and promise.

### Suggestions for improvement

Adding summary statistics for the main paper on the standard benchmarks would be very helpful. There is likely room for something that small. And part of Fig 6 could be moved if not.

---

> ### Author Response · Authors · 2022-07-26
> **Author Response to Reviewer BArk**
>
> Thank you for your feedback! Your summary is accurate. Below, we respond to your concern about the time steps mentioned in the intro and we emphasize the significance of our paper. Could you please let us know whether this addresses your concerns or whether there are still remaining issues we could address?
>
> > The evaluation falls short of the initial promise in the intro. To the knowledge of the reviewer, none of the tasks studied have the length of time horizon promised in the intro. Thus it remains unclear if this method indeed works better in that promised setting. The proposed method is clearly helpful in settings with sparse rewards, so perhaps that should be the motivation and promise.
>
> Thank you for pointing out this potential misunderstanding. We wrote "complex control problems can require millions of time steps" just to motivate hierarchical RL. The sparse reward tasks in our paper require several hundreds of time steps (although discovering the rewards in the first place takes longer). We'll add a sentence to the intro that explicitly says this.
>
> > Adding summary statistics for the main paper on the standard benchmarks would be very helpful. There is likely room for something that small. And part of Fig 6 could be moved if not.
>
> That's a great idea! The camera-ready version allows for an extra page, so we will move the full results of the standard benchmarks into the main text.
>
> We would like to emphasize that previous hierarchical RL methods have not been able to solve a wide range of pure RL tasks (no pretraining tasks, demonstrations, semantic goal space, etc) while ensuring that the hierarchy was used (no task reward given to the low level). Director not only learns successful hierarchical behaviors across a wide range of environments, it further learns them directly from pixels. We thus think that Director constitutes a significant step forward for hierarchical RL research.

---

> ### Author Response · Authors · 2022-08-07
> **Discussion**
>
> Dear Reviewer BArk,
>
> The discussion period is coming to an end soon and we haven't received a response from you yet. Could we please ask you to confirm whether our response has resolved your concerns or whether you see any remaining issues that motivate your current rating? If there are remaining issues, we would be more than happy to address them or further clarify where necessary.
>
> Thank you!

---

### Official Review · Reviewer_uwa5 · 2022-07-14

**Rating:** 5
**Confidence:** 4
**Soundness:** 3 good
**Presentation:** 4 excellent
**Contribution:** 3 good

**Summary:**

In this paper, the authors introduce a learnable RL-based planner. It is specifically design for long horizon sparse reward environment. The proposed planner consists of two major components. A world model uses an off-the-shelf encoder to represent the raw pixel input and additional goal autoencoder is trained to obtain a more sparse world representation. On the other hand, a manager policy and a worker policy are trained to predict the next goal and the next atomic action. The authors also designs exploration reward and goal reward to stimulate the agent. Experiments are conducted on public benchmarks including long horizon navigation and atari games.

**Questions:**

Q1: The authors may want to explain why the "manager" policy can be well trained with the async-trained world model decoder from a more theoretical perspective. To the best of my understanding, the autoencoder is trained by memory replay but the "manager" is on-policy. Given the fact the the world model is trained separately, how to guarantee that the "z" in world encoder share the same space with the "z" in the manager? The paper will be much stronger if there is a theoretical justification.

Q2: There are two rewards to stimulate the agent. One is for exploration and the other one is for reaching the goals. Do these two rewards compete with each other? For example, the reward of exploring a goal state is larger than reaching it. How to avoid it?

Q3: What is the magic behind changing the goal very k=8 step?

Q4: Why Dreamer performs so bad in Ant Maze XL and Pin Pad Six?

Q5: Any insights for the reason that the proposed method is weaker in Atari games?

I've been away from this community for a while, therefore I will also consider other reviewers comments. I will rise my rating if the authors can provide insightful responses. Based on the submitted manuscript, I lean to accept.

**Limitations:**

See weakness.

**Strengths And Weaknesses:**

Strength:
The overall presentation for the proposed method and experiments is friendly for audience to understand although there are some minor grammar issues. It is a challenging problem for RL agent to plan in long horizon and reward-sparse environment. According to some of the experiment results, the proposed method works very well (Ant Maze). The idea of using a more sparse representation to represent abstract action is interesting and the overall design of the proposed method is reasonable.

Weaknesses:
Most of the components used in the proposed method are off-the-shelf, but I don't think it is a big concern regarding the originality. The design of the "manager" mechanism is not very solid. See question below. Although the proposed method show promising results in some tasks, in general RL benchmarks it is slightly weaker than the popular baselines. It will be better to discuss more about it.

---

> ### Author Response · Authors · 2022-07-29
> **Author Response to Reviewer uwa5 (Part 2/2)**
>
> > Q3: What is the magic behind changing the goal very k=8 step?
>
> The imagination rollouts of the world model have a length of 16 steps. We a goal twice during the rollout, at steps t=0 and t=8. This results in a goal duration of K=8. We also experimented with K=4 and K=16 and found that they result in similar performance, with shorter durations benefitting fast-paced environments and longer durations benefitting long-horizon tasks with sparse rewards. We will add the experimental results for this ablation to the appendix of the final paper. We see researching mechanisms for dynamic goal durations as an interesting future direction, for example based on a heuristic of whether a goal has been achieved.
>
> > Q4: Why Dreamer performs so bad in Ant Maze XL and Pin Pad Six?
>
> These environments feature sparse rewards that require exploration techniques beyond a stochastic policy to discover within the provided interaction budget. Dreamer performs poorly on these tasks because it fails to discover the reward.
>
> - The only reward in Ant Maze is given when the ant touches the goal object, so from the initialized position the ant has to discover how to locomote and navigate all the way to the other side of the maze before receiving its first reward.
> - In the Pin Pad environments, the only reward is given once the agent steps on all pads in exactly the correct sequence.
>
> > Q5: Any insights for the reason that the proposed method is weaker in Atari games?
>
> Thank you for this question, we will add insights on this to the paper. "Pure" Director underperforms Dreamer on fast-paced Atari games because of their need for precise, fast movement. The worker receives no task reward to learn these fast movements, instead it has to be steered by the manager through goals. However, the goals only change every K=8 steps, which is too slow for quickly moving back and forth in state-space. A simple remedy is presented in the same figure, where we give task reward to the worker policy, so that it can learn to perform fast motions near the current goal that serve the task, without the manager having to communicate those detailed motions. An interesting future direction is to find heuristics to switch goals when the previous goal is reached, which could be an elegant solution here.

---

> ### Author Response · Authors · 2022-07-29
> **Author Response to Reviewer uwa5 (Part 1/2)**
>
> Thank you for your thoughtful review! We added clarification of the manager training and the way rewards are combined, added insights about the performance on Atari games, and responded to your remaining questions. Please let us know whether this fully addresses your concerns or whether there are any points remaining, which we would be happy to address.
>
> > Q1: The authors may want to explain why the "manager" policy can be well trained with the async-trained world model decoder [...] how to guarantee that the "z" in world encoder share the same space with the "z" in the manager? The paper will be much stronger if there is a theoretical justification.
>
> There is probably a small misunderstanding here. If the explanation below does not answer your question, please let us know and we'll be happy to clarify further. All components in Dreamer and Director are optimized concurrently to improve throughout the agent's lifetime --- by "separately" we just mean that gradients are stopped between components.
>
> As we understand your question, it is more a question about Dreamer than Director, namely why we can continuously optimize both (1) the world model and (2) the policy that takes world model representations as inputs. The answer is that both components are optimized throughout training, so as the world model representations change the policy is trained to adapt to the changes. In Dreamer (and Director), the policy is optimized on batches of size 16K time steps through massively parallel rollouts, so it can adjust quickly to changes in the world model.
>
> We also point out that "z" in our paper refers not to the world model representations but to the latent space of the goal autoencoder that learns on top of the world model representations. The manager's actions stay aligned with the latent space of the goal autoencoder because the manager treats both the goal autoencoder and the worker as a "black box" --- it simply chooses 8x8 discrete actions that maximize future task rewards and exploration rewards. As such, it adapts it's strategy as the goal autoencoder and worker policy change over time.
>
> > Q2: There are two rewards to stimulate the agent. One is for exploration and the other one is for reaching the goals. Do these two rewards compete with each other?
>
> In Director, the manager maximizes both task reward and exploration reward, whereas the worker maximizes only the goal reward. Thus, it is the task and exploration rewards that are being combined, not the exploration and goal rewards. The only exception is "Director (worker task reward)" in Appendix A.
>
> To combine multiple reward signals, we use the following mechanism described in Section 2.3. We use the imagined rollout to compute two separate returns for the two reward signals, divide each by its exponential moving standard deviation, and then sum them with weights w^extr = 1.0 and w^expl = 0.1. This ensures that the extrinsic/task reward contributes stronger to the policy gradient than the exploration bonus. As a result, the agent explores in the absence of rewards but once rewards are found, the agent pays more attention to them than to the exploration signal.
>
> (continued below...)

---

> ### Author Response · Authors · 2022-08-07
> **Discussion**
>
> Dear Reviewer uwa5,
>
> The discussion period is coming to an end soon and we haven't received a response from you yet. Could we please ask you to confirm whether our response has resolved your concerns or whether you see any remaining issues that motivate your current rating? If there are remaining issues, we would be more than happy to address them or further clarify where necessary.
>
> Thank you!

---

### Meta-Review · Area_Chair_HJbs · 2022-08-25

**Recommendation:** Accept
**Confidence:** Less certain

**Metareview:**

This paper studies an interesting problem, and overall the reviewers agreed the exposition and validation are sufficient. We encourage the authors to consider the issues raised by the reviewers and further improve the work in the final version.

**Award:**

No

---

### Decision · Program_Chairs · 2022-09-14

Accept